# Variation in appendages in early Cambrian bradoriids reveals a wide range of body plans in stem-euarthropods

Dayou Zhai[1,2], Mark Williams [2,3], David J. Siveter[2,3], Thomas H.P. Harvey[2,3], Robert S. Sansom[4], Sarah E. Gabbott[2,3], Derek J. Siveter[2,5,6], Xiaoya Ma[1,2,7], Runqing Zhou[8], Yu Liu[1,2] & Xianguang Hou[1,2]

Traditionally, the origin and evolution of modern arthropod body plans has been revealed through increasing levels of appendage specialisation exhibited by Cambrian euarthropods. Here we show significant variation in limb morphologies and patterns of limb-tagmosis among three early Cambrian arthropod species conventionally assigned to the Bradoriida. These arthropods are recovered as a monophyletic stem-euarthropod group (and sister taxon to crown-group euarthropods, i.e. Chelicerata, Mandibulata and their extinct relatives), thus implying a radiation of stem-euarthropods where trends towards increasing appendage specialisation were explored convergently with other euarthropod groups. The alternative solution, where bradoriids are polyphyletic, representing several independent origins of a small, bivalved body plan in lineages from diverse regions of the euarthropod and mandibulate stems, is only marginally less parsimonious. The new data reveal a previously unknown disparity of body plans in stem-euarthropods and both solutions support remarkable evolutionary convergence, either of fundamental body plans or appendage specialization patterns.

---

[1] Yunnan Key Laboratory for Palaeobiology, Yunnan University, 650091 Kunming, Yunnan, China. [2] MEC International Joint Laboratory for Palaeobiology and Palaeoenvironment, Yunnan University, 650091 Kunming, Yunnan, China. [3] Centre for Palaeobiology Research, School of Geography, Geology and the Environment, University of Leicester, Leicester LE1 7RH, UK. [4] School of Earth and Environmental Sciences, University of Manchester, Oxford Road, Oxford M13 9PT, UK. [5] Earth Collections, Oxford University Museum of Natural History, Parks Road, Oxford OX1 3PW, UK. [6] Department of Earth Sciences, University of Oxford, South Parks Road, Oxford OX1 3PR, UK. [7] Centre for Ecology and Conservation, College of Life and Environmental Sciences, University of Exeter, Penryn Campus, Penryn, Cornwall TR10 9FE, UK. [8] Institute of Geology and Geophysics, Chinese academy of Sciences, 19 Beituchengxi Road, 100029 Beijing, China. Correspondence and requests for materials should be addressed to Y.L. (email: yu.liu@ynu.edu.cn) or to X.H. (email: xghou@ynu.edu.cn)

The bivalved bradoriid euarthropods are a prolific and geographically widespread group in Cambrian to Lower Ordovician strata worldwide[1]. They are reported as the most abundant euarthropods in the Burgess Shale and Chengjiang Lagerstätten[1,2] and clearly formed an important component of Cambrian marine benthic and nektobenthic communities. Most bradoriid species are known only from their carapace. Based solely on the observation that bradoriids are small and have a bivalved carapace (adults typically range up to about 5 mm long), more or less all such fossils in the Cambrian and Lower Ordovician have traditionally been assigned to the Bradoriida, and formerly they were, *ipso facto*, regarded as the earliest evidence of ostracod crustaceans (Archaeocopida[3]).

Previous analyses of incomplete soft anatomies from two bradoriid species from the exceptionally preserved Chengjiang biota of Yunnan Province, China, namely *Kunmingella douvillei* (Mansuy, 1912)[4] and *Kunyangella cheni* (Huo, 1965)[5], recovered a monophyletic Bradoriida as stem-crustaceans[6] or as the sister group to total group Mandibulata[7]. Here we describe new material of three bradoriid taxa, *K. douvillei*, *K. cheni* and *Indiana* sp., from the Chengjiang biota. As revealed by micro-computed tomographic (micro-CT) scanning, these bradoriids preserve unprecedented three-dimensional (3-D) detail of their soft anatomy revealing hitherto unknown and marked morphological differences in limb anatomy between the species. This fundamentally poses the question of whether the

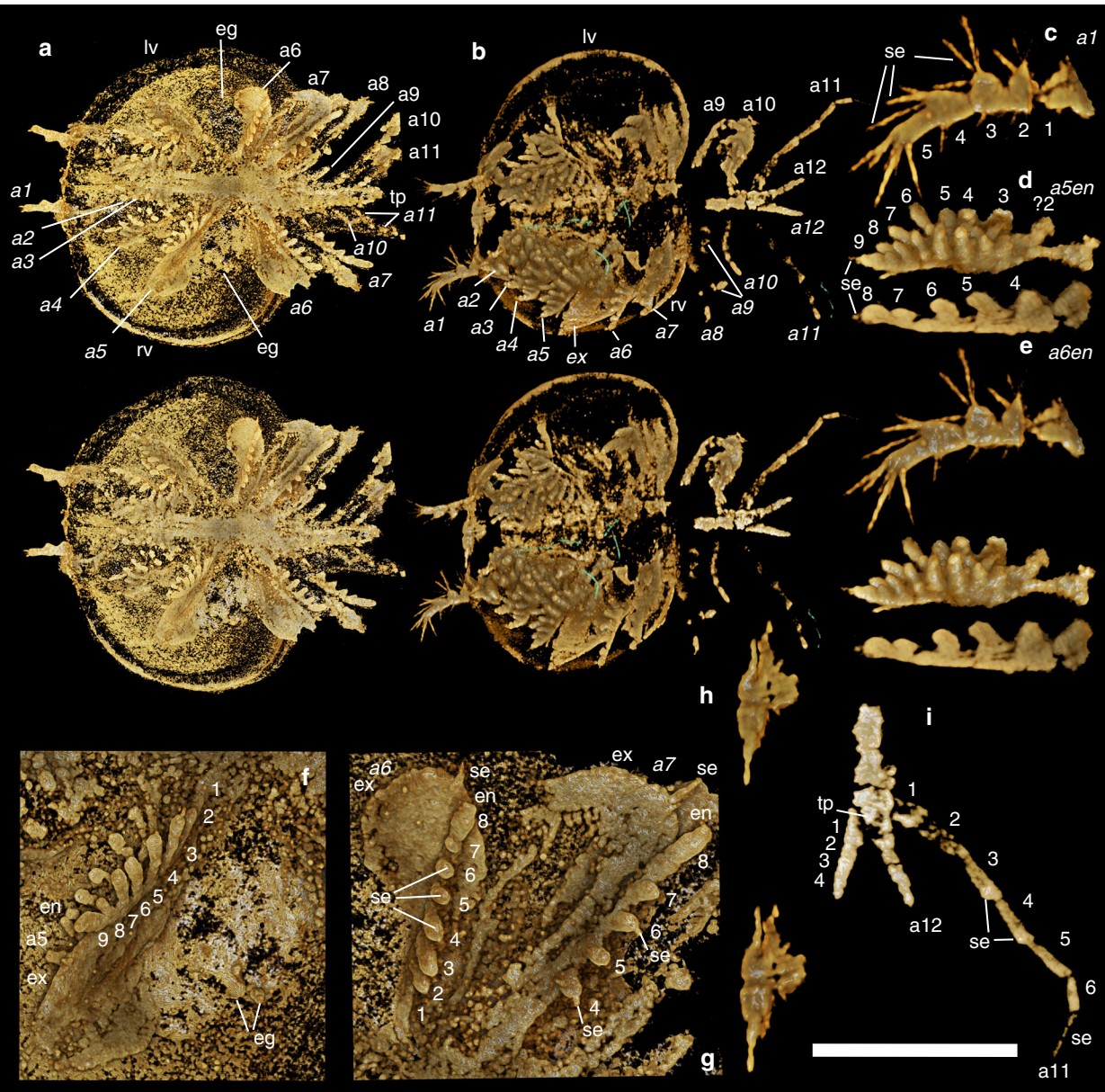

**Fig. 1** *Kunmingella douvillei* (Mansuy, 1912): **a**, **f**, **g** (YKLP 16235); **b**–**e**, **h**, **i** (YKLP 16233). All views are ventral. **a**–**e**, **h** Stereo-pairs with a 20° tilt. **a**, **b** Whole animal views: in **a**, eggs can be seen mid-valve, nestled in the region of the lobal structure; in **b**, the green coloured elongate structures appear to be organic but are not part of the bradoriid. **c** Right appendage 1. **d** Endopod of right appendage 5 (incomplete proximal part). **e** Endopod of right appendage 6 (proximal podomeres not shown). **f** Right appendage 5. **g** Left mid and some posterior trunk appendages. **h** Possible neural structures. **i** Posterior-most part of the trunk and limbs. Scale bar: **a** 2.19 mm; **b** 1.65 mm; **c** 560 μm; **d** 850 μm; **e** 610 μm; **f** 820 μm; **g** 730 μm.; **h** 680 μm; **i** 810 μm. a1–a12 1st to 12th appendage (italic signifies a right-side appendage), eg egg(s), en endopod, ex exopod, se seta(e), tp tailpiece, numbers 1–9 refer to the podomeres and/or endites, from distal to proximal

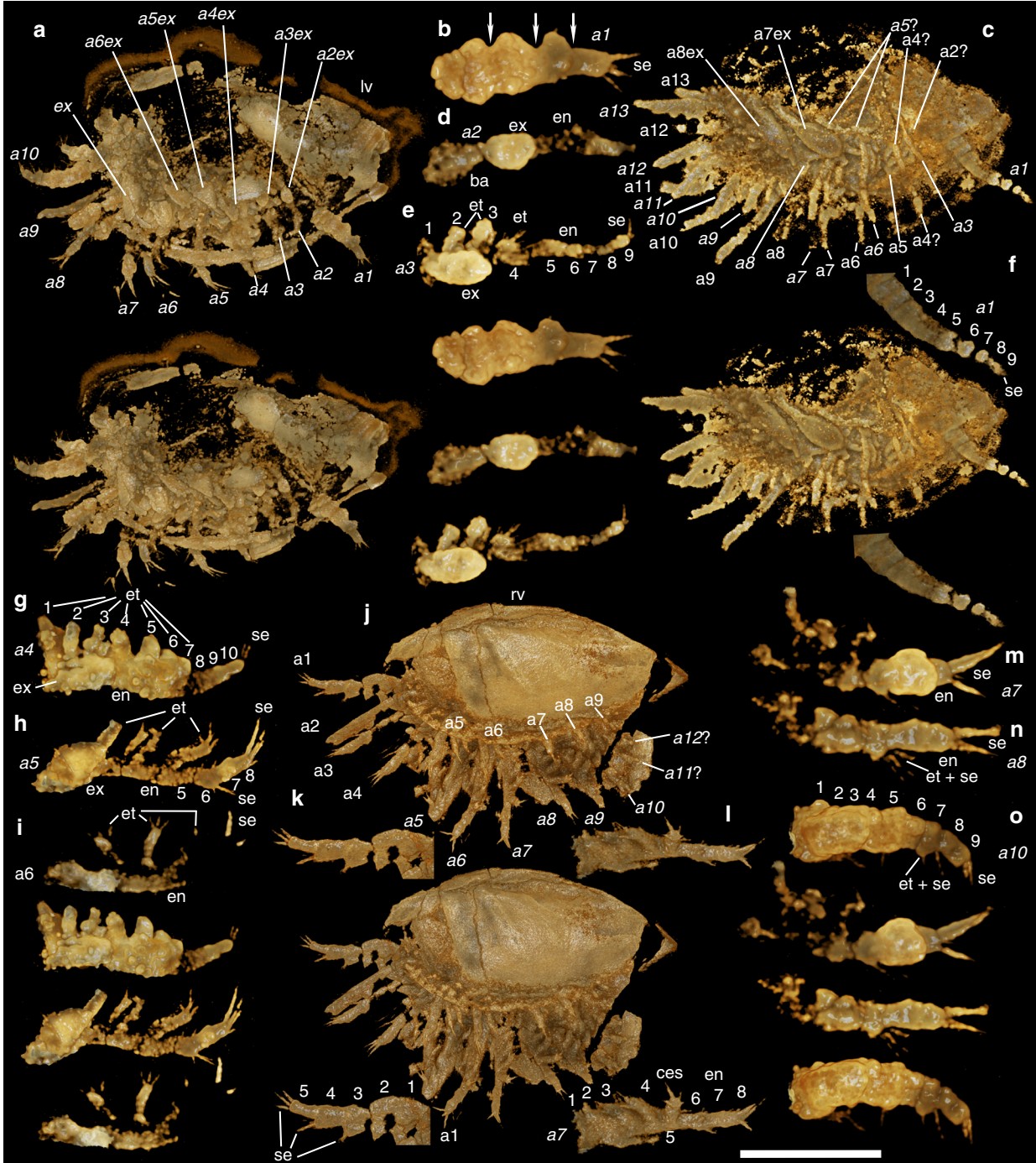

**Fig. 2** *Kunyangella cheni* (Huo, 1965) and *Indiana* sp.: **a–i**, **m–o** *Kunyangella cheni* (**a**, **b**, **d**, **e**, **g–i**, **m–o** YKLP 16232; **c**, **f** YKLP 16236); **j–l** *Indiana* sp. (YKLP 16231). All images are stereo-pairs with a 20° tilt. **a** Right lateral view. **b** Incomplete right appendage 1, right lateral view, showing four distal podomeres (arrows mark podomere boundaries). **c** Right lateral view. **d** Right appendage 2, right lateral view. **e** Right appendage 3, right lateral view. **f** Right appendage 1, right lateral view. **g** Right appendage 4, right lateral view. **h** Right appendage 5, right lateral view. **i** Left appendage 6, right lateral view. **j** Left lateral view. **k** Left appendage 1, left lateral view. **l** Right appendage 7, left lateral view. **m** Right appendage 7, right lateral view. **n** Right appendage 8, right lateral view. **o** Right appendage 10, right lateral view. Scale bar: **a** 1.20 mm; **b** 430 μm; **c** 1.38 mm; **d** 390 μm; **e** 490 μm; **f** 990 μm; **g** 570 μm; **h** 640 μm; **i** 570 μm; **j** 960 μm; **k** 610 μm; **l** 560 μm; **m** 390 μm; **n** 380 μm; **o** 460 μm. Abbreviations additional to Fig. 1: ba basipod, ces castellate endite with seta(e), et endite (s), numbers 1–10 refer to the podomeres and/or endites, from distal to proximal. For *Indiana* sp., the full complement of setae on podomeres 2–4 of the antennae (a1) are visible in Supplementary Fig. 2

Bradoriida is a non-monophyletic group. Alternatively, they may be a monophyletic group but one that represents an independent radiation of stem-euarthropods where trends towards increasing appendage specialisation were developed convergently with other groups.

## Results

**Micro-CT analysis of soft anatomy.** Using micro-CT scanning, we herein reveal remarkable details of soft anatomy that demonstrate that bradoriids constitute an assemblage of highly disparate euarthropods with substantially more diverse limb morphologies

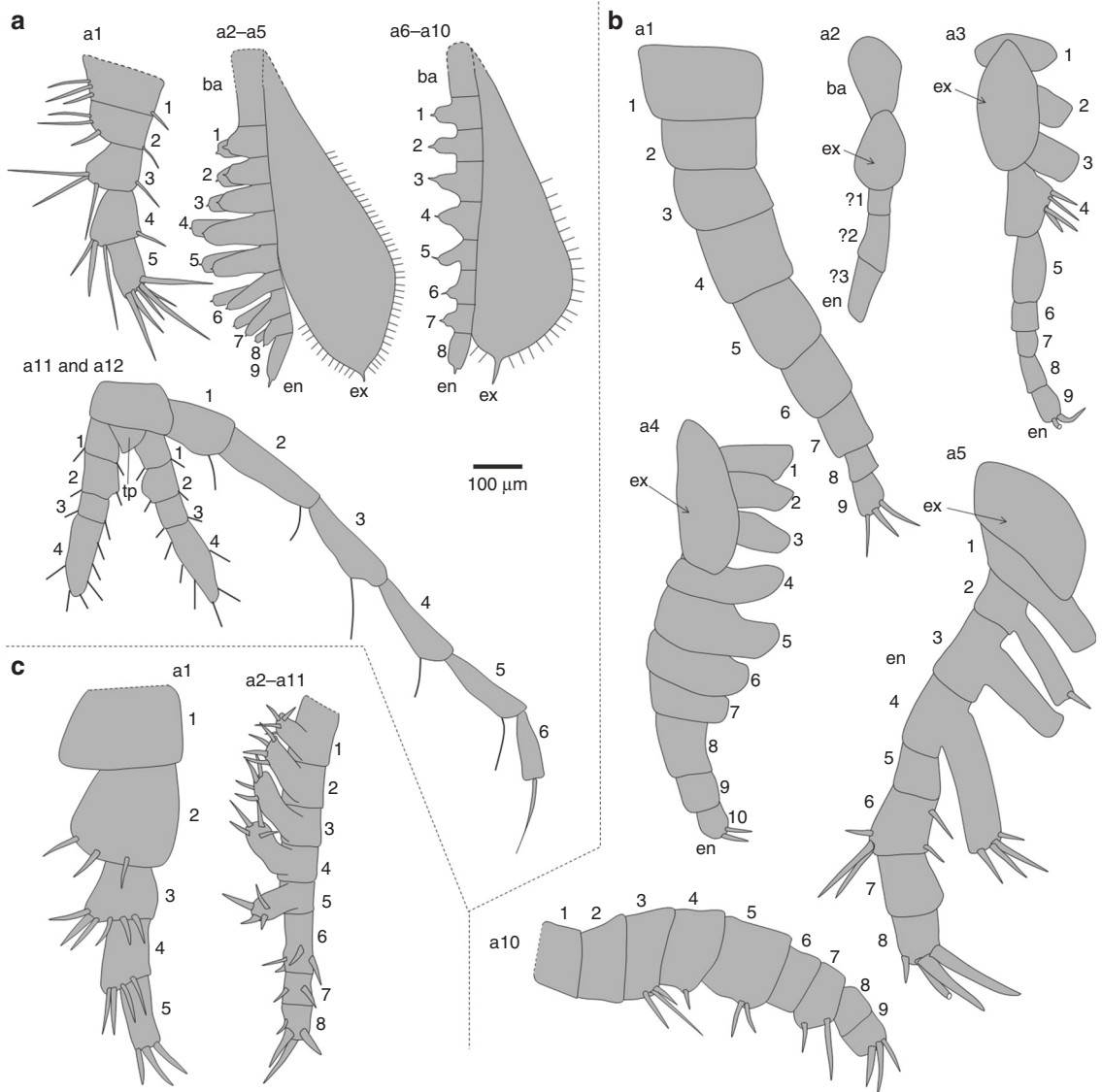

**Fig. 3** Limb morphology based on Figs. 1 and 2. **a** *Kunmingella douvillei*. **b** *Kunyangella cheni*. **c** *Indiana* sp. Abbreviations as for Fig. 1. For *K. douvillei* the exopod morphology of appendages a2 and a3 are inferred, as is the basipod morphology for a2–a10. Both terminal appendages are shown (a12), in order to show their context to the tailpiece (tp). For *Indiana* sp., the distribution of setae on podomeres 2–5 of the antenna (a1) is based on Fig. 2k and Supplementary Fig. 2

than previously known (Figs. 1–3, Supplementary notes). Thus *K. douvillei* exhibits 12 appendages with a distinction between an anterior set of 4 post-antennular appendages with double rows of endites and a posterior set of five appendages with single rows of endites, as well as distinct anterior-most and posterior-most uniramous appendages (Figs. 1 and 3). *K. cheni* has a succession of four head appendages that are each distinct from one another, followed by nine trunk appendages, the posterior-most appendage being uniramous (Figs. 2, 3). *Indiana* sp., however, possesses a morphologically distinct antenna followed by a homonomous series of 11 uniramous appendages (Figs. 2 and 3). Our interpretations of the anatomy of the bradoriid appendages are taphonomically informed, with effects of decay distinguishable from primary absence of features (see Supplementary Fig. 1).

**Phylogenetic analysis**. To test whether bradoriids represent a polyphyletic or monophyletic group, we conducted a cladistic analysis of *K. douvillei*, *K. cheni* and *Indiana* sp. using the phylogenetic database of Siveter et al.[8] (Fig. 4), which has the

taxonomic coverage necessary. In the most parsimonious solution, the three bradoriid taxa consistently formed a monophyletic clade as sister taxon to crown-group euarthropods (Chelicerata, Mandibulata and their extinct relatives). This contrasts with previous analysis of this data set, which found Bradoriida to be stem-mandibulates[7] or stem-crustaceans[6]. A monophyletic Bradoriida as recovered in our analysis implies extensive character evolution within the group that is convergent on trends seen elsewhere in the euarthropod tree. To explore the full implications of this result, we conducted a survey of limb differentiation along the body (*sensu*[9]), which provides a comparative measure of specialisation along the appendage series, without requiring knowledge of appendage homologies[10,11]. *Indiana* sp. exhibits an extremely low degree of limb specialisation with a limb formula of (1,11), i.e. a morphologically distinct antenna followed by a homonomous series of 11 appendages. This degree of limb differentiation is otherwise restricted to certain lobopodians, anomalocaridids and 'short-great appendage' arthropods (see ref. [11]). In contrast, *K. douvillei* exhibits a distinction between an

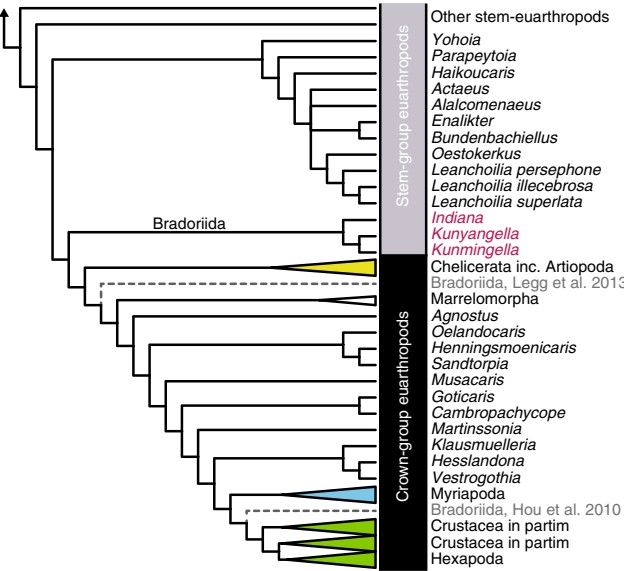

**Fig. 4** Revised affinity of the three Bradoriida taxa (*Indiana* sp., *Kunmingella douvillei* and *Kunyangella cheni*) as sister taxon to the euarthropod crown-group. Strict consensus of the most parsimonious trees resulting from an implied weighting ($k = 3$) parsimony analysis following refs. [8,9]. Former interpretations of Bradoriida[7,8] are included. Bradoriida monophyly is supported by the presence of a free head shield and rows of elongate spines on the inner margin of the frontal appendage (characters 22 and 145, respectively, both homoplastic). Bradoriida are united with the crown-group euarthropods on the basis of their unspecialised or antennate third head segment, contrasting with other stem-euarthropods, which are interpreted as having megacheiran-style appendages (character 188). They are not placed within the euarthropod crown-group because they have eight or more podomeres in their locomotory limbs (contrasting with five or six +terminal claw in most crown-euarthropods; character 280)

anterior set of appendages with double rows of endites and a posterior set with single rows, as well as distinct anterior-most and posterior-most uniramous appendages, i.e. a limb formula of (1,4,5,1,1), with no obvious correspondence in other euarthropods. *K. cheni* is different again, with a succession of four head appendages that are distinct from one another, giving a limb formula of (1,1,1,1,8,1), more similar to mandibulates (i.e. modern pancrustaceans plus various Cambrian 'Orsten' taxa).

## Discussion

Traditionally, increasing levels of appendage specialisation have been associated with the establishment of crown-group euarthropod body plans, particularly in Pancrustacea/Mandibulata[12,13]. Recently, fossil evidence has demonstrated unanticipated levels of appendage specialisation and body tagmosis in the euarthropod stem, including among lobopodians[11]. In this sense, bradoriids could add to a picture of complex transitions (experimentation) in stem-group euarthropod body plans. To test the alternative hypothesis, that bradoriids are not monophyletic and limb transitions were fewer, we ran a constrained parsimony phylogenetic analysis to minimise the multiple appendage specialisations required by the bradoriids (i.e. *Indiana* sp. constrained to a position on the euarthropod stem closer to the root than *Fuxianhuia*; *K. cheni* constrained to have a position within the mandibulates closer to crustaceans than *Marrella*; and *K. douvillei* left unconstrained). Bayesian analyses to evaluate alternative topologies were not possible given the lack of convergence in this data set. In our constrained parsimony analysis (unfigured), *Indiana* sp. resolved

with other bivalved stem-euarthropods (such as *Nereocaris* and *Odaraia*) and *K. cheni* resolved within crustaceans, while *K. douvillei* variously resolved with either of the other bradoriid taxa or between them. The most parsimonious constrained trees were only marginally less parsimonious than the unconstrained trees (lengths of 141.99 and 141.16, respectively), meaning it is not possible to rule out this alternative on the basis of the available data. In this polyphyletic scenario, bradoriids would represent several independent origins of a small, bivalved body plan in lineages from diverse regions of the euarthropod and mandibulate stems and an astounding case of convergence from disparate origins.

Whichever of our two hypotheses is correct, it is now clear that a miniaturised, bivalved body form arose early on in arthropod evolution, among stem-euarthropods with comparatively unspecialised limbs (particularly so in *Indiana* sp.). At the same time, other stem-euarthropods (notably *Isoxys*) were exploring macroscopic bivalved body forms. Both required extreme modification from the plesiomorphic arthropod body plan, and both were rediscovered by comparatively derived mandibulates (phosphatocopines, phyllocarids, some branchiopods and ostracods). Bradoriids therefore add to a picture of early innovation combined with complex transitions in stem-euarthropod body plans.

## Methods

**Materials**. The bradoriids were collected from the Yu'anshan Member, Chiungchussu Formation, *Eoredlichia-Wutingaspis* trilobite biozone, Cambrian Series 2, Stage 3, Yunnan Province, China. Two specimens of *K. douvillei* (YKLP 16233, YKLP 16235) were collected from Mafang, Haikou, Kunming; one specimen of *K. cheni* (YKLP 16232) is preserved on the same rock slab with *K. douvillei* YKLP 16233. A second *K. cheni* (YKLP 16236) was collected from Erjie, Jinning, Kunming. The specimen of *Indiana* sp. (YKLP 16231) was collected from Mt. Jianshan, Haikou. All specimens are from event beds[14]. The bradoriids are laterally compressed but with a degree of 3-dimensionality (valve thickness up to about 90 μm, overall compacted thickness of the animals up to about 410 μm) which has been emphasised in our figures by using stereo-photography with a tilt angle of 20° (Figs. 1 and 2). The appendages of any one specimen show different degrees of compression (e.g. *K. cheni* YKLP 16232, Supplementary Fig. 1), although none of the appendages are as systematically compressed as those fossils preserved as black carbon films in the so-called background beds of the Chengjiang Lagerstätte[14]. The mineralogical composition of the fossils is similar to that previously reported for Chengjiang specimens[15–17]. Iron oxides pseudomorphing framboidal or microcrystalline pyrite coat the fossil surface, resulting in a high-density contrast between fossil and matrix and facilitating micro-CT imaging. The preservation of an organic film, also known from Chengjiang fossils, cannot be resolved through micro-CT imaging owing to a lack of density contrast between organic film and matrix and/or that the organic film is too thin to be distinguished. However, scanning electron microscopy energy-dispersive X-ray analysis (*K. cheni* YKLP 16232) has revealed a very weak carbon signal in some areas.

**Photography**. Fossil structures that are exposed on the surface of the rock slabs were imaged with a Leica M205C stereo-microscope.

**Micro-CT**. Fossil structures unexposed, within the slabs, were scanned using a Zeiss Xradia 520 Versa X-ray Microscope (used for YKLP 16231, YKLP 16235 and YKLP 16236) and a GE Phoenix Nanotom M X-ray scanner (used for YKLP 16232 and YKLP 16233). The two instruments produced comparable results. Scanning resolution ranged from 3.0 to 6.1 μm, depending on the size of the scanned region and the slab. The data from each specimen, in the form of a series of TIFF images (one to a few thousand in number), representing cross-sections through different parts of the slab, were processed with the Drishti (Versions 2.4 and 2.6.4) software to generate 3-D models of the fossils figured herein (Figs. 1 and 2, Supplementary Fig. 1).

**Phylogenetic analysis**. Phylogenetic analysis utilised the database of Siveter et al.[8]. We have revised the character codings for *K. cheni* and *K. douvillei* in light of our new data and have added *Indiana* sp. Furthermore, character 23 (bivalved carapace) was revised so as to be contingent on the presence of a free head shield (character 22; for an explanation of this character, see also ref. [18], characters 25 and 26). The data set was analysed using parsimony analysis with implied weighting ($k = 3$, following ref. [7]).

**Reporting summary**. Further information on research design is available in the Nature Research Reporting Summary linked to this article.

## Data availability
Data are available from the authors on request. Fossil specimens are deposited at the Yunnan Key Laboratory for Palaeobiology (YKLP), Yunnan University, China, with the accession numbers: YKLP 16231, 16232, 16233, 16235, 16236. For the phylogenetic analysis depicted in Fig. 4, the data matrix and associated search commands are available via MorphoBank, project 3499 (http://morphobank.org/permalink/?P3499).

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

## Acknowledgements
This study is supported by NSFC grants 41861134032 and 2015HC029 and Yunnan Provincial Research Grants 2018FA025 and 2018IA073. Shaogang Zang, Shuyan Hou, Xiaoyu Yang and Chunjie Cao assisted with micro-CT and SEM analyses. Ajay Limaye (Australian National University), Jing Lu and Yuzhi Hu (Institute of Vertebrate Paleontology and Paleoanthropology, Chinese Academy of Sciences) provided guidance with Drishti software. M.W. thanks the Leverhulme Trust for a Research Fellowship (RF-2018-275/4). We are grateful for the constructive comments of the three reviewers. This is publication number 2019002 from the University of Leicester Centre for Palaeobiology Research.

## Author contributions
X.H. collected the specimens. D.Z., Y.L. and X.H. designed the research. D.Z., Y.L. and R.Z. scanned the specimens and processed micro-CT data. D.Z., M.W., David J. Siveter, T.H.P.H., Derek J. Siveter and X.M. interpreted the soft anatomy. David J. Siveter and D.Z. prepared the figures. M.W., David J. Siveter and D.Z. wrote the first draft of the manuscript with scientific and editorial input from all authors. R.S.S. conducted phylogenetic analysis with input from M.W., T.H.P.H. and D.Z. S.E.G. and D.Z. examined taphonomy.

## Additional information

**Competing interests:** The authors declare no competing interests.

