## [Peer Review File · Communications Biology]

Reviewers' comments:

Reviewer #1 (Remarks to the Author):

Please see attachment below

Reviewer #2 (Remarks to the Author):

The data presented in this study are very nice and for sure should be published. Also the interpretations provided by the authors are in general reasonable, though I have some points of criticism, especially concerning the alternative hypothesis (see below). In general, I consider this a minor to moderate revision, detailed comments are below.

Title: After having read the entire manuscript, I think that the current title is too far-fetched. I do not see the real "challenge" which the authors claim to arise from the new data. Better rephrase.

33: Bradoriida is a group and hence singular. Better write "numerically abundant representatives of Bradoriida"; also check in other instances

36ff: I think I understand the sentence, but it is not well formulated. Suggestion: "The alternative solution, in which there is no monophyletic group Bradoriida, but several independent origins of this morphotype from ancestors with a small, bivalved body..." or something similar

41f: What is the difference between "evolutionary convergence" and "convergences from disparate origins"? I do not see any. Even if it will be explained further below, the reader might be confused here.

50: "presumed adults typically range up to about 5 mm length" (not long)

51: There should be no "the" before group names as they are singular: to Bradoriida; also check in other instances

55: Is it journal style to only cite the species describer without year? Very unusual to me.

57: Just say species, not taxa, when it is all about species here

62: a polyphyletic group would (sensu Hennig) mean that the group has erroneously been interpreted as a monophyletic group as convergencies have been interpreted as synapomorphies. At this point I would phrase more general: "whether Bradoriida is in fact no monophyletic group"

62: no question mark here

72: post-antennular would be more precise than post-antennal

83: Please differentiate between the abbreviations of the two different genera starting with K.

99, 102: Always also use the species name, not just the genus name. Also check in the remaining text.

104f: You can use mandibulates in this way, but you should also cite a reference here and maybe add that this group was called Crustacea sensu lato by other authors

112: Is polyphyly the only alternative hypothesis? Would paraphyly also be an option? Or would the alternative hypothesis simply be that the group is non-monophyletic, without deciding between poly- and paraphyly? I think the authors should be clearer here (and earlier, see above), and probably the last option would be the best one.

Figures:

I find it very unfortunate to have one scale bar for all panels on one figure, which has then very weird sizes. Better have one scale bar for each panel with proper sizes (e.g. 2 mm instead of 2.19 mm). If there is not sufficient space, I would recommend to put the content of one figure onto two and make the different panels larger.

It would be better to have the stereo-pairs left and right of each other, not top and bottom, as the reader currently has to turn the page (or the computer) to see the effect.

Fig. 2: Not sure what is meant with "castellate". Maybe rephrase?

Fig. 3: The drawings of the appendages should better be slightly idealised. Currently, the lines shake a bit too much.

Fig. 4: As far as I understood the text, it is not entirely clear if Bradoriida is monophyletic or not. However, Fig. 4 has a clear monophyletic result. It would be more honest to also show a tree with non-monophyletic Bradoriida.

Best wishes,
Carolin Haug, LMU Munich

Reviewer #3 (Remarks to the Author):

This paper presents a very beautiful fauna of early forms of Euarthropods. This Bradoriida fauna presents an exceptional preservation of appendages.

This paper is very done and could be published in state with some very small corrections:

Line 57 replace "describe" by "present"

It is a shame that earliest papers on Chengjiang bradoriid appendages are not cited in the main text body (Hou, Siveter et al. 1996 is cited in supplementary file and Shu, Vannier et al 1999 is not cited)

I am not specialist of phylogenetic analysis so I can't judge the reproductibility of the data given in supplementary file

Our responses to the reviewers' comments are given in red

Reviewer #1 (Remarks to the Author):

It is a good writing manuscript. The application of the Micro-CT scanning to the research of Burgess Shale-type fossils is a remarkable finding. Dr. Zhai and Dr. Liu, as well as their colleague, they open a new window to see the 3-D preserved animal, especially for the arthropods with detailed preserved appendage structures, which normally covered by the carapace or imbedded by the rock. Bradoriids are so common in Early Cambrian marine system, but the information of soft-bodied parts is really rare. It is good to see the soft anatomy of the bradoriids and the monophyletic result, at least for the three analyzed taxa. The manuscript showed the limb tagmosis various in three bradoriids, and made further comparisons of the head limb specialization between Bradoriida taxa and some euarthropods, such as the mandibulate stems. If all of the observation and the understanding by Dr. Zhai and his colleague are correct, thus the disparity and the degree of limb differentiation of bradoriids have been underestimating for a long time. The results in this manuscript will change our idea of the stem-group arthropod evolution. Before the manuscript published, please double check the “limb formula” of the *Kunmingella* and *Kunyangella*.

1) For the first taxon *Kunmingella*, the anterior set of first five appendages is different from the posterior set of appendages in bearing the two-rowed endites. However, when we focus on the best preserved anterior appendage, that is a5 (Fig. 1f), we can see the endopod is just half of the exopod in length. Clearly, the reconstruction showed us the endopod and the exopod are equal in length (Fig. 3a, see a2-a5). So, it is reasonable to believe that the two-rowed endite bearing branch actually represents the proximal part of the endopod. In contrast, the posterior set of appendage with single row of endites, best preserved in a6 (Fig. 1a, 1e) sound only seen in the distal part. Accordingly, it is possible that the post-antenna appendages consist of a proximal part with double rows of the endites, and a distal part with single row of endites. Additionally, the endopod is of the double rows of the endites is not common in Cambrian arthropods, only see in the frontal appendage of Radiodontants in my knowledge.

We provide a range of images to support our interpretations of limb morphologies. Interpretation of the endopodal morphologies of *Kunmingella douvillei* are based on two specimens (Fig. 1a, 1b), rather than only on the dissected a5 (Fig. 1d) and a6 (Fig. 1e). For a2-a5, it is clear in Fig. 1b that the two-rowed endites extend from the basal part of the endopod to the penultimate podomere. For a6-a10, at least 8 podomeres with single-rowed endites are seen (e.g., Fig. 1a, 1g, left a6). Where the endopod seems to be slightly shorter than the exopod (e.g. right a5, Fig. 1f), this is because of the curvature of the endopod. In other appendages of this specimens (Fig. 1a) the endopod and exopod are subequal in length.

2) For the second taxon *Kunyangella*, there are two specimens have been scanned, the YKLP16236 and YKLP 16232. In the former, no detail morphology of the appendages could be seen, except for a1 (Fig. 2C). In the latter, the author proposed the head section with four specialized head appendages a1-a4 (Fig. 2a). However, a1, a2 and a3 are poorly preserved, so it is hard to say if they are distinct from one another (Fig. 2b,d,e). So I have to doubt on the understanding of the “high degree of limb specialization” and “the similarity to the head tagmosis of mandibulates” (p. 5, line 4-12) in *Kunyangella*. But the criteria are just based on the 2-D images in the manuscript, maybe we can see more clear images of a1-a4 based on their 3D results.

Reviewer 1 is correct in that 3D results give a better understanding of the limb morphologies than the 2D images. Indeed, we have presented images as stereo-pairs for this very purpose. Although the distal part of a2 (Fig. 2d) is poorly preserved, the differences between a1-a4 are nonetheless significant. Firstly, for a1, in both YKLP 16232 (Fig. 1a, 1b) and YKLP 16236 (Fig. 1c, 1f) this is uniramous; a2 does not possess endites; a3 is well preserved (as evidenced by the preservation of fine setae) and has 4 endites on the proximal endopod podomeres whilst the distal 5 have no endites; a4 is also well preserved, with 7 endites, whilst the distal 3 podomeres have no endite but setae are present on the last podomere. The preservation of fine setae and the continuous outline of the limbs indicate that the effect of taphonomic bias is minor and that the morphologies presented basically represent the true nature of the specimens.

3) It is interesting that the different number of the body appendages in the two specimens of *Kunyangella*. But it is always difficult to say the adjoining limbs are the left-right ones, or the antero-posterior ones. Such as the number of the juvenile *Kumingella* in Hou et al., 2010. Fortunately, we can solve this problem by the scanning 3D image. It will be very helpful to see the specimens of *Kunyangella* and may be the juvenile of *Kumingella* in the ventral view.

In addition to the stereo-pairs presented herein, we have checked the appendages in the 3D model generated by Drishti software. In Fig. 1a and Fig. 1c, we have annotated both the left and the right appendages; we are confident that we have not misidentified the order of the limbs.

4) The distinction between the head and the trunk of the *Kumingella* and *Kunyangella* is not evidence-based. The definition of the head appendage, in my opinion mainly based on the limb specialization, rather than the limb anterior extension (some research followed the definition by Hou, 2010). Please see Fig. 1b, the trunk appendage a6 and a7 also extended anteriorly. Why don't we assign them into the head part?

We show a clear head-trunk distinction as evidenced by the different morphologies of the appendages, rather than solely on their orientation.

5) I strongly recommend to reformat the figures 1 and 2. Put the labels in the same position if possible. Both the images with and without the symbols sound unnecessary.

Figures 1 and 2 represent a substantial body of work in themselves and have been painstakingly assembled mindful to show the full range of morphology with the minimal of necessary labelling. We do not wish to reformat them.

Some minor mistakes:

1) Fig.1a and Fig. 1b were overlapped on a10;

No overlap is evident.

2) Fig. 3a, either show two pairs of limbs, a11 and a12 in both sides, or just show them in one side.

We have modified the figure caption to note that both of the terminal appendages for a12 are shown. This is important, to give context to these limbs relative to the terminal piece.

3) Fig.1f, “a5” should be *italic*

We have corrected this (please note that only where the appendages such as ‘a5’ etc. are distinguished have we used italics)

4) Fig. b, a1, lack of the distal claw

We consider that this comment relates to Fig. 3b, where we have now added 3 distal setae (based on YKLP 16232, see Fig. 2b).

Reviewer #2 (Remarks to the Author):

The data presented in this study are very nice and for sure should be published. Also the interpretations provided by the authors are in general reasonable, though I have some points of criticism, especially concerning the alternative hypothesis (see below). In general, I consider this a minor to moderate revision, detailed comments are below.

Title: After having read the entire manuscript, I think that the current title is too far-fetched. I do not see the real “challenge” which the authors claim to arise from the new data. Better rephrase.

We have modified the title to ‘Appendages in early Cambrian bradoriids reveal a wide range of body plans in stem-euarthropods’.

33: Bradoriida is a group and hence singular. Better write “numerically abundant representatives of Bradoriida”; also check in other instances

With respect, Bradoriida is being considered as a taxon here and therefore the grammar is correct.

36ff: I think I understand the sentence, but it is not well formulated. Suggestion: “The alternative solution, in which there is no monophyletic group Bradoriida, but several independent origins of this morphotype from ancestors with a small, bivalved body...” or something similar

We have modified the sentence to the following:

“The alternative solution, in which bradoriids are polyphyletic and represent several independent origins of a small, bivalved body plan in lineages from diverse regions of the euarthropod and mandibulate stems, is only marginally less parsimonious.”

41f: What is the difference between “evolutionary convergence” and “convergences from disparate origins”? I do not see any. Even if it will be explained further below, the reader might be confused here.

We have modified the sentence to the following:

“In all instances, the new data reveal a previously unknown disparity of body plans in stem-euarthropods and both solutions support a remarkable case of evolutionary convergence, either of fundamental body plans or appendage specialization patterns.”

50: “presumed adults typically range up to about 5 mm length” (not long)

Either is grammatically and technically correct here, so 5 mm long is OK

51: There should be no “the” before group names as they are singular: to Bradoriida; also check in other instances

With respect, it should be ‘the Bradoriida’, as in this case it is referring to a particular taxon, and therefore the definite article should be used.

55: Is it journal style to only cite the species describer without year? Very unusual to me.

This was done in order to avoid duplication where the reference was cited at the point where the author was quoted. We have now added the dates where necessary.

57: Just say species, not taxa, when it is all about species here

The word ‘taxa’ is used very specifically here, in that we are referring to two named species and one taxon that is referred to in open nomenclature, ‘Indiana sp.’

62: a polyphyletic group would (sensu Hennig) mean that the group has erroneously been interpreted as a monophyletic group as convergencies have been interpreted as synapomorphies. At this point I would phrase more general: “whether Bradoriida is in fact no monophyletic group”

We have rephrased as suggested.

62: no question mark here

The question mark is considered to be correct, as given.

72: post-antennular would be more precise than post-antennal

We have changed the text to post-antennular.

83: Please differentiate between the abbreviations of the two different genera starting with K.

We have used “*Km. douvillei*” and “*Ky. cheni*”, to differentiate.

99, 102: Always also use the species name, not just the genus name. Also check in the remaining text.

We have corrected this.

104f: You can use mandibulates in this way, but you should also cite a reference here and maybe add that this group was called Crustacea sensu lato by other authors

Within the manuscript we cite Legg et al. 2013 with regard to the definition of the total group Mandibulata (and of the crown group). Cross-reading of Figure 4c in that paper will immediately indicate that the total group Mandibulata is

equivalent to Crustacea *sensu lato* of other authors (e.g., as used by Zhang et al. 2012, *Gondwana Research* 21, 1115-1127).

112: Is polyphyly the only alternative hypothesis? Would paraphyly also be an option? Or would the alternative hypothesis simply be that the group is non-monophyletic, without deciding between poly- and paraphyly? I think the authors should be clearer here (and earlier, see above), and probably the last option would be the best one.

Paraphyly is possible, though unparsimonious (step-wise evolution of limb tagmosis patterns through a grade of bradoriids, then switching to something else without a bradoriid-like body plan). Intuitively this would be less parsimonious than either monophyly or polyphyly. Here we use constraint trees to explicitly test the possibility that limb appendage specialization patterns are more conservative which by necessity means we test a polyphyletic solution for Bradoriida (not paraphyletic). As such our manuscript is asking whether it's more likely that a small, bivalved body morphology evolved more than once, or whether diverse patterns of limb tagmosis evolved within a bradoriid clade. It's less parsimonious that we have the latter combined with a loss of bradoriid-like body form, under a paraphyletic scenario. Therefore, we have left the text as it is.

Figures:

I find it very unfortunate to have one scale bar for all panels on one figure, which has then very weird sizes. Better have one scale bar for each panel with proper sizes (e.g. 2 mm instead of 2.19 mm). If there is not sufficient space, I would recommend to put the content of one figure onto two and make the different panels larger.

We thank the referee for this suggestion, which we have carefully examined, but consider that several scales are unnecessary and, moreover, would lead to more disruption of morphological information than it will provide. It is standard practice in countless palaeontological papers to have (as we have) only one scale for the several panels of a figure. Adding several scales to our Figures 1 and 2 is practically impossible (space constraints). Furthermore, in no way do we wish to lose the integrity of the information we provide in each figure by disassembling the 2 figures into 3 or more (Figs 1 and 2 have been painstakingly assembled mindful of the morphological comparisons being made).

It would be better to have the stereo-pairs left and right of each other, not top and bottom, as the reader currently has to turn the page (or the computer) to see the effect.

We have used top to bottom stereo-pairs as this maximizes the amount of information on the figures' moreover, it has the advantage that the specimens

are viewed in life orientation in lateral view. There is a long history of using this style for bivalved arthropods (see *A Stereo-Atlas of Ostracod Shells*).

Fig. 2: Not sure what is meant with “castellate”. Maybe rephrase?

Castellate means ‘turreted, castle-like’, alluding to the shape of the endites, Its use is correct.

Fig. 3: The drawings of the appendages should better be slightly idealised. Currently, the lines shake a bit too much.

Thank you. We have followed this suggestion.

Fig. 4: As far as I understood the text, it is not entirely clear if Bradoriida is monophyletic or not. However, Fig. 4 has a clear monophyletic result. It would be more honest to also show a tree with non-monophyletic Bradoriida.

We present the most parsimonious result as a figure which is monophyly of Bradoriida. The alternative constrained version (polyphyly for Bradoriida) is poorly resolved and presenting it as a figure would add little more information than is presented in the text.

Reviewer #3 (Remarks to the Author):

This paper presents a very beautiful fauna of early forms of Euarthropods. This Bradoriida fauna presents an exceptional preservation of appendages.

This paper is very done and could be published in state with some very small corrections:

Line 57 replace "describe" by "present"

The word ‘describe’ is correct, as that is what we do in the manuscript.

It is a shame that earliest papers on Chengjiang bradoriid appendages are not cited in the main text body (Hou, Siveter et al. 1996 is cited in supplementary file and Shu, Vannier et al 1999 is not cited)

Shu *et al.* 1999 is indeed cited, though in the supplementary file. We have cited the earlier literature there because this is where we deal with the systematic palaeontology.

I am not specialist of phylogenic analysis so I can't judge the reproductibility of the data given in supplementary file

The phylogenetic analysis utilised the dataset of Siveter *et al.* 2017 and is in the public domain.

REVIEWERS' COMMENTS:

Reviewer #1 (Remarks to the Author):

I am satisfied with the responses and the revision given by the authors. Look forward to seeing the manuscript published in Communications Biology.

Reviewer #3 (Remarks to the Author):

Very nice paper which could be published in the present state